# Effect of Blade Curvature on Fan Integration in Engine Cooling Module

**Manuel Henner \*, Bruno Demory, Mohamed Alaoui, Maxime Laurent and Benjamin Behey**

Valeo Thermal System, 8, rue Louis Lormand, CS 80517 La Verrière, 78322 Le Mesnil Saint Denis, France; bruno.demory@valeo.com (B.D.); mohamed.alaoui@valeo.com (M.A.); maxime2.laurent@valeo.com (M.L.); benjamin.behey@valeo.com (B.B.)

**\*** Correspondence: manuel.henner@valeo.com

**Abstract:** Two blade curvatures representative of those found in automotive fans are compared. Measured performances are analyzed for forward and backward curved blades, either with or without heat-exchangers placed in front of them. The backward fan demonstrated good efficiency but poor acoustics, whereas it is the contrary for the forward fan. Investigations are completed by a numerical analysis of the flow in the cooling module. Different integration effects are highlighted depending on the blade curvature, showing variation in pressure, torque and efficiency. Analyses of blade loadings show that the flow is more homogeneous with a forward curved fan and it produces less unsteadiness at the blade tip. Post-processing of detached eddy simulations (DES) shows density fluctuations on the blade wall and confirms the correlation between the large vortical structures and the acoustic sources for both fans. In addition, with the forward fan, the sound propagation is less directed towards the axis of rotation and it yields up to −3.6 dB of sound pressure level (SPL) measured in front of the cooling module. As a conclusion, any choice for a fan must result from a compromise between aerodynamics and aeroacoustics, and the final performances must be carefully checked on the module.

**Keywords:** fan; fan system; aeroacoustics; cooling module; blade curvature

---

## 1. Introduction

Automotive thermal management relies strongly on the cooling module that is placed on the front-end of the car. It is composed of a set of coolers and a fan system, both designed to maximize heat exchanges in various conditions and minimize the acoustic annoyance. Stringent new regulations on vehicle efficiency and noise emission have more than ever yielded investigations of the multi-physic optimization of the fan, assessing performance in the actual context, i.e., once integrated in the cooling module.

Fan parameterization has been therefore widely used to conduct fan optimization, and the most frequent approaches are based on the use of a design of experience (DoE) with geometrical parameters [1,2]. Several attempts were also made to introduce a multi-physics modelling of response surfaces, either for mechanical concern (i.e., fan deflexion and modal analysis [3]) or for acoustics [4].

It is widely recognized that the blade curvature along the blade span is among the most important parameters to be accounted for in acoustics. Several studies have already explained some differences in fan behavior according to their design [5–8]. All the observations show the importance of the load at the blade tip since the flow interacts with the recirculation in the tip clearance and with the unsteady phenomenon thus created.

Engineers who design fans face several difficulties when sizing them. First of all, they have to make compromises of all kinds between aerodynamic performance, acoustics, packaging requirements,

etc. The choices that must be made are all the more difficult as it is not easy to anticipate their consequences when the fan is finally mounted into its cooling module, i.e., when the fan has to operate behind a radiator and in the center of a very compact shroud. Due to the integration effects that vary from one geometry to another, it is ultimately uncertain whether a fan system optimized for bench measurement conditions is the most appropriate for the thermal cooling of a vehicle.

## 2. Objective

The objective of the present work is to study and understand the phenomena that occurs when a fan is placed behind a stacking of heat exchangers. It is common to observe that the performance in a cooling module is not equal to the sum of the pressure drop of the radiator and the pressure increase through the fan. Discrepancy comes from additional losses in the cooling module that are due to the integration effect.

Integration effects may vary depending on the fan type, i.e., whether it is designed with backward or forward curved blades. This feature is frequently used either to increase aerodynamic performance or to improve acoustics, and is a technical choice that is taken a priori when designing the fan. A good understanding of the consequences of blade curvature on the performance of the cooling module therefore requires knowledge of the loss mechanisms, both to minimize them and to correctly dimension the components. The challenge is also to make informed trade-offs between aerodynamic and aeroacoustic performance.

## 3. Cooling Module Description and Methods for Fan Comparison

Cooling modules must meet the requirements of modern vehicles for which compactness is essential. This is difficult to achieve because several heat exchangers are required: they are most frequently composed of a radiator, a condenser for air conditioning, sometimes a charge air cooler, and oil coolers or even some supplementary heat exchangers for the electrical vehicles equipped with a heat pump.

### 3.1. The Shroud

Cooling modules are subject to space constraints in the motor compartments, and this often results in asymmetrical shapes. It is the case for the module used in this study and presented in Figure 1. The two different fans that can be fitted into the shroud are presented in the next paragraph. The shroud is used to hold the motor and fan unit and to ensure the aerodynamic convergence of the air passing through the exchangers and going to the fan. It is a nearly flat surface with a maximum depth of 36 mm. Therefore it cannot be considered as a profiled convergent (like in a nozzle), but rather as a sealing device.

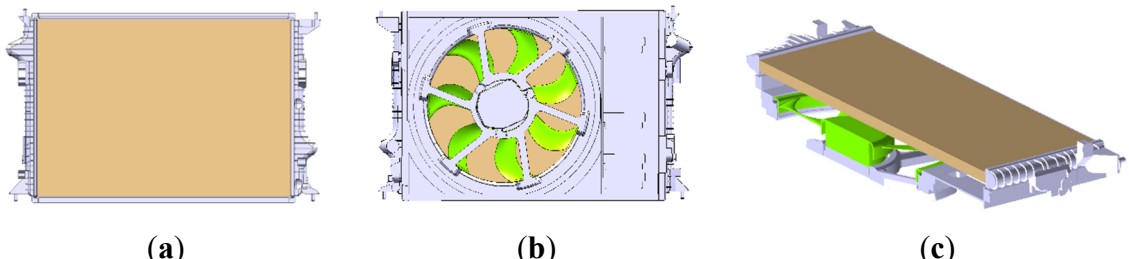

|         (a)         |         (b)         |         (c)         |

**Figure 1.** Overview of a cooling module: (**a**) front view, (**b**) rear view, and (**c**) radiator and fan system (cut view).

The arms supporting the motor are arranged a few mm (16 mm) from the trailing edge of the fan. They therefore participate by their potential effects in the tonal noise generation, mainly the blade passage frequency (BPF) and the 1st and 2nd harmonics (H1 and H2).

For the purposes of this study, this shroud is kept identical in the comparison of the two different fans.

### 3.2. Fan Systems

In general, for the automotive fan, the blade stacking from the bottom to the top is made in such a way that the trajectory of the trailing edges at all radii traces a flat surface. The curvature is given by orienting the blade either forward (in the direction of rotation) or backward. It is therefore a combination of sweep and lean changes, which are in theory a stacking change either in the direction of the chord or perpendicular to the chord [9,10].

These curvature modifications give very different and even antagonistic characteristics. It is frequently observed that for automotive operating points (high flow rate with a high specific speed $N_s$ of 3 to 4, and a low specific diameter $d_s$ of 1 to 1.5) rear curves allow reaching higher flow rates, unfortunately at the price of a higher acoustics level [4,7,11].

For fans, specific speed and diameter are calculated from the formula

$$N_s = \frac{\Omega \cdot \sqrt{Q}}{\Delta P^{\frac{3}{4}}} \tag{1}$$

$$d_s = D \cdot \Delta P^{\frac{1}{4}} \cdot Q^{\frac{1}{2}} \tag{2}$$

with $\Delta P$: pressure rise (Pa), Q: volumic flow rate (m$^3$/s), D: diameter (m), and $\Omega$: rotational speed (rad/s).

Two fans are therefore considered in this study (Figure 2). They have been selected according to their curvature characteristics, which are opposite to each other, i.e., a backward (BW) and a forward curvature (FW). Both have 7 blades equally distributed. The diameter at the blade tip is 423 mm, and 162 mm for the hub. The chords go from 42 to 59 mm on the backward fan, and from 67 to 86 mm on the forward fan. It is important to note that parameters such as chord, blade thickness and stagger angles have been intentionally modified to have fans with comparable performances. This could not have been the case otherwise since reversing the curvature leads to significant performance changes.

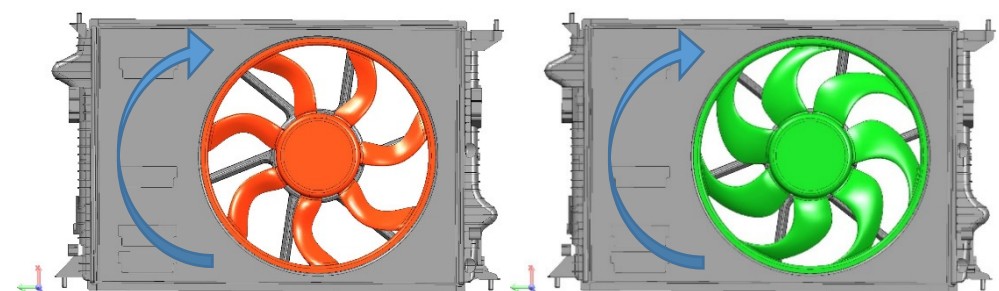

**Figure 2.** Backward and forward curved blades (blue arrows: rotational direction).

It is important to note that these fans are equipped with a rotating ring which has two functions: on the one hand, to maintain the end of the blades because they are made of plastic and would otherwise undergo significant deformation; and on the other hand to limit the recirculation effects between the blade tip and the shroud [12]. This ring also has an L-shaped profile, which both increases its rigidity and allows better control of the recirculating air flow. For mechanical reasons in the event of shock or vibration, and also for reasons of variability in the process, the peripheral tip clearance is quite large (4.8 mm at the radius, i.e., ~2% of the radius) and the recirculating flow can reach 6% of the nominal flow rate.

### 3.3. Experimental Facility

In order to investigate the integration effect of the fan in the cooling module, the two fans are tested in the same shroud described above by placing or removing the radiator (comparison of the fan with or without the heat exchanger). In order to ensure the tightness of the device, possible micro-leaks between the shroud and the radiator core were eliminated during tests by applying adhesive tape all around the cooling module. In addition, great attention was paid to the manufacturing quality of the fans that have been machined from aluminum. They do not deform under the effect of rotation (a known phenomenon on plastic fans) and they do not have imperfections due to the molding process, which sometimes leaves burrs on the parting surfaces of the mold.

Each device (BW fan with or without exchanger, FW fan with or without exchanger) was tested on the aerodynamic bench at the La Verrière R&D site (France). This bench meets ISO Standard DP 5801 [13] and was presented in [14]. The static pressure is measured as the difference between the atmospheric pressure and the static pressure in the bench plenum (see Figure 3). The performance curve is obtained for a constant fan rotation speed by varying the flow rate on the bench using two calibrated nozzles. The fan is driven in rotation by an electric motor whose consumption is known by measuring voltage and amperage. Its efficiency was measured on a dynamometer device for the electrical motor, which makes it possible to estimate the mechanical torque produced.

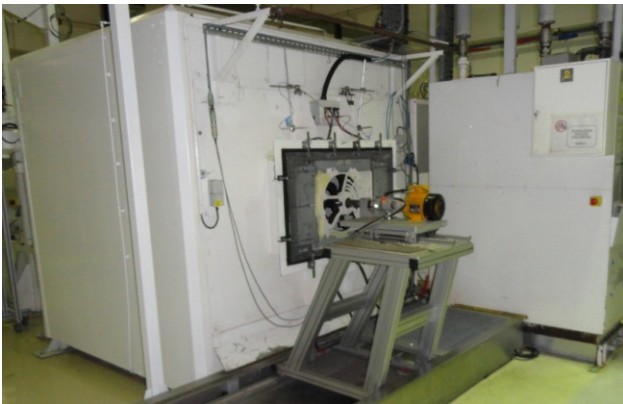

**Figure 3.** Test facility for the fan system or module performance measurement.

At the end of the aerodynamic tests, acoustic measurements were taken in a semi-anechoic chamber at the nominal rotational speed of each of the fans (Figure 4). Measurements were taken on the axis of rotation of the fan, at a distance of 1 m from the front face of the fan hub and at a height of 1 m. The cooling module is suspended accordingly by flexible fasteners to filter out vibrations.

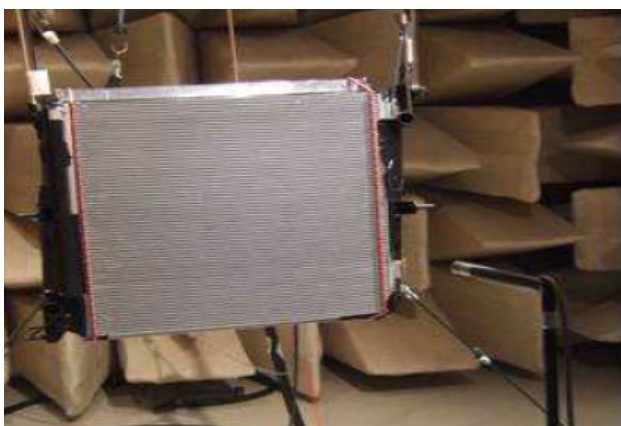

**Figure 4.** Semi-anechoic chamber, with the cooling module hanged and microphone positioning.

*3.4. Numerical Simulation*

In order to better understand the differences in fan behavior with or without radiators, numerical simulations of the flows were carried out. Their purpose was to compare the blade loading and analyze the flows. The commercial software used (CCM+) uses a polyhedral mesh and the simulation domain includes the cooling module and the external geometry of the test bench (see Figure 3). The mesh size includes 45 million polyhedra, mainly concentrated on the fan and its supporting arms (about 80%). A great deal of attention is given to taking into account the boundary layers, since the value of Y+ is very close to 1 (less than 2 on the whole fan except the leading edges, where it is between 2 and 4). Correlation studies give an accuracy of less than 5% error on overall performance [13] for fan cases.

The simulations were conducted according to a protocol starting with a stationary calculation (RANS), and supplemented by an unsteady calculation (URANS), with a two equations turbulence model (k-$\omega$ SST). Convergence was checked by monitoring residues and global flow quantities. In general five fan rotations are required to achieve a low numerical residue level and to obtain a stabilized solution.

The study is completed by detached eddy simulations (DES), with a time step of $5 \times 10^{-6}$. The acoustic estimates are based on an acoustic analogy with free-field propagation, using namely the FW&H model [15]. Only dipole sources are considered when using the walls of the fan and the shroud as emitting surfaces. This analytical model of free field propagation is valid for compact sources, which is the case if we consider the low chord length of the blades compared to the wave lengths (chord length ~5 cm, wave length for the BPF frequency ~1 m). On the other hand, reflection, scattering and diffraction effects of acoustic waves and the acoustic attenuation of exchangers are not considered. In the case of automotive fans, the low number of Mach (Mach 0.2) indicates that the sources are purely dipole and consequently come from pressure fluctuations on the walls. Since the sound pressure level is several orders of magnitude lower than the aerodynamic pressure level, the DES calculations are performed with the compressible model in order to obtain the best accuracy.

## 4. Comparative Performance of Fans with or without Heat Exchanger

*4.1. Comparative Performance of Fans with Heat Exchanger*

The operating point is set according to the thermal performance of the heat exchangers that must evacuate a fixed amount of calories. Taking into account their characteristics, the air flow rate must be 3053 m$^3$/h. If we refer to the pressure drop curve of the cores measured with a homogeneous air flow, the pressure to be supplied by the fan would be 140 Pa.

For the two fans considered, the aerodynamic performance is obtained in the complete cooling module for rotational speeds of 2404 and 2615 rpm, respectively, for the fans BW and FW (see performance curves in Figure 5). At this operating point, the pressure increase given by the fan perfectly balances the pressure losses, and the upstream and downstream pressures are equal to zero (zero pressure point). This should correspond to the condition for which a vehicle is at rest and does not benefit from the dynamic effect of air when the vehicle is running. In reality, the additional pressure drop of the air intake (grid, vehicle front end) and engine compartment is not taken into account in this case, which is why the operating point incorporates a certain safety margin. Fortunately, this additional pressure drop is quickly erased as soon as the vehicle moves forward, and at high speed the cooling module generally resists air with a fan that ends up being a turbine (i.e., it is the fluid that gives its energy).

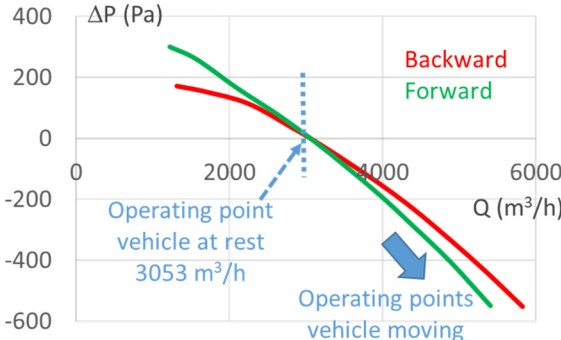

**Figure 5.** Performances in the cooling module.

In order to produce a fair comparison, the effects on fan efficiency should also be considered. It is calculated by the ratio between aerodynamic power (product of flow rate and pressure) and mechanical power (product of torque and speed). The difficulty lies in the fact that it is not possible to make a precise evaluation of the static pressure produced by the fan in the module, and therefore the theoretical pressure drop is considered, neglecting additional pressures losses due to the integration or the flow inhomogeneity.

Efficiency is therefore calculated using the formula:

$$Eff. \, (\%) = \frac{\Delta P \cdot Q}{T \cdot \Omega} \tag{3}$$

with T: torque (Nm).

At the nominal operating point, the pressure rise is considered to be 140 Pa (the theoretical value from the radiator characteristics).

It is then noted that these performances are obtained with different efficiencies (31% and 28.2% in global efficiency respectively for BW and FW with a motor efficiency of ~80%, or 38.9% and 35.3% for the mechanical efficiency) and therefore the BW fan consumes 60 Watt less electricity (−11%).

*4.2. Comparative Performance of Fans without Heat Exchanger*

For the purposes of the study, these two fans were tested in a configuration without a heat exchanger for the rotational speeds with which they had been validated in the complete module. Their performances are presented in Figure 6.

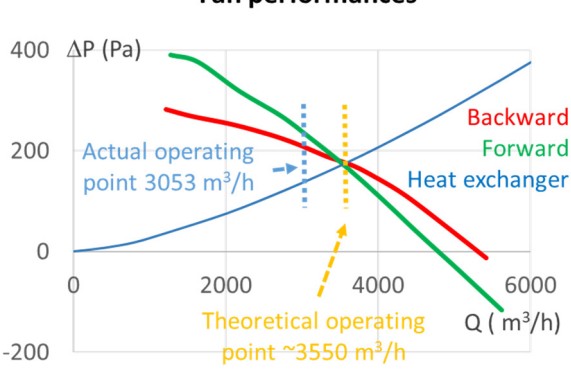

**Figure 6.** Fan performances compared to the radiator resistance.

It can be seen that the two fans exhibit different behaviors, characterized by the fact that the BW blades produce less pressure at low flow rates, but that the maximum flow rate (zero pressure

point) is higher. This change in the slope of the performance curve is particularly important because it is observed that at the expected nominal operating point (3053 m³/h), the pressure delivered by the forward curvature is in this case higher than that of the rear curvature (231 versus 206 Pa), but its efficiency is still lower (46.7% versus 49.9%, see performance Table 1).

**Table 1.** Compared performances of BW and FW fans, with or without the heat exchanger.

|  | Flow Rate (m³/h) | ΔP Module Theoretical (Pa) | ΔP Module Simulation (Pa) | ΔP Fan (Pa) | Torque in Module (N.m) | Torque Fan (N.m) | Efficiency Module (%) | Efficiency Fan (%) |
|---|---|---|---|---|---|---|---|---|
| BW | 3053 | 140 | 174 | 206 | 1.51 | 1.39 | 38.9 | 49.9 |
| FW | 3053 | 140 | 169 | 231 | 1.55 | 1.53 | 35.3 | 46.7 |

This comparison highlights that it is difficult to know which fan to select for a cooling module if you only have its performance on a bench, without any information on integration effects. If a theoretical approach were to be followed by comparing the fan curves and the pressure drop curve of the radiator, the adaptation points would be estimated at 3575 and 3536 m³/h respectively for the backward and forward curvatures (in Figure 6 it is indicated by the average value of 3550 m³/h).

### 4.3. Integration Effect on Overall Performances

It is possible at this stage to quantify the integration effects by comparing the ideal pressure drop of the exchangers with the true pressure variation created by the fan with or without exchangers. One can also note that fans undergo different variations in their performances according to the type of curvature (see Figure 7). The backward curvature shows a difference of 66 Pa (206 Pa without the exchanger compared to the ideal pressure drop of 140 Pa of the radiator), while it is slightly higher for the forward curvature with 91 Pa (231 Pa without the exchanger compared to the 140 Pa of the radiator).

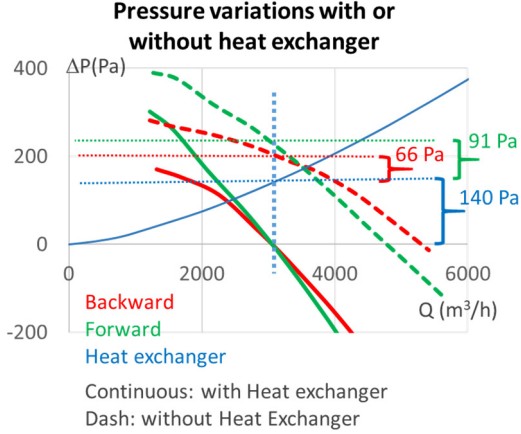

**Figure 7.** Differences between theoretical pressure losses in the heat exchanger and observed pressure variations by integration in the cooling module.

It should be noted that the forward blade undergoes practically no torque variation (1.55 vs. 1.53, respectively, with and without the exchanger, i.e., ~1%), while it is largely modified for the backward curvature (1.51 vs. 1.39 respectively with and without the exchanger, i.e., ~9%). Regarding combined effects of both pressure and torque variations, the static efficiency changes from fan alone to the module configuration are about the same for the two fans, i.e., −11% and −11.4%, respectively, for the backward and the forward curvatures (see Figure 8).

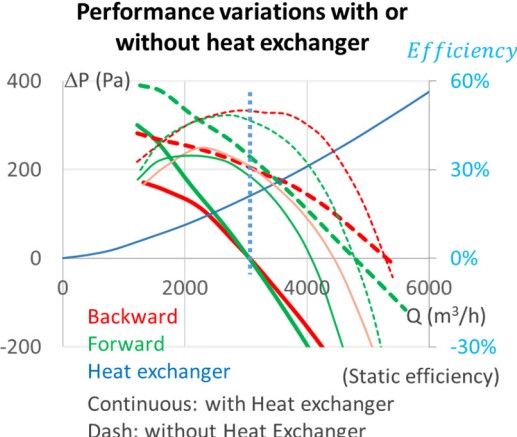

**Figure 8.** Comparison of the fan system global efficiency (calculated with the fan system pressure rise) to module efficiency (calculated with the theoretical pressure losses in the heat exchanger). Electrical motor efficiency is ~80%.

### 4.4. Fan Acoustics

The choice of one fan or the other is not limited to considering its airflow performance, since acoustics play an increasingly important role in the compromise to be found and several mechanisms compete in the noise generation [16]. The measurements made give the spectra of Figure 9 where the contribution of broadband noise can be observed, with more pronounced low frequencies (when using the backward blades) and higher pre-harmonic humps before H1. The latter is weaker on the forward curved blades. On the other hand, it is observed that the broadband noise sees a small hump between 3000 and 4000 Hz with the forward fan. This has been investigated and presented in [17] with a strong suspicion of a Tollmien-Schlichting wave. The phenomenon is much weaker here and caution must be exercised in this hypothesis.

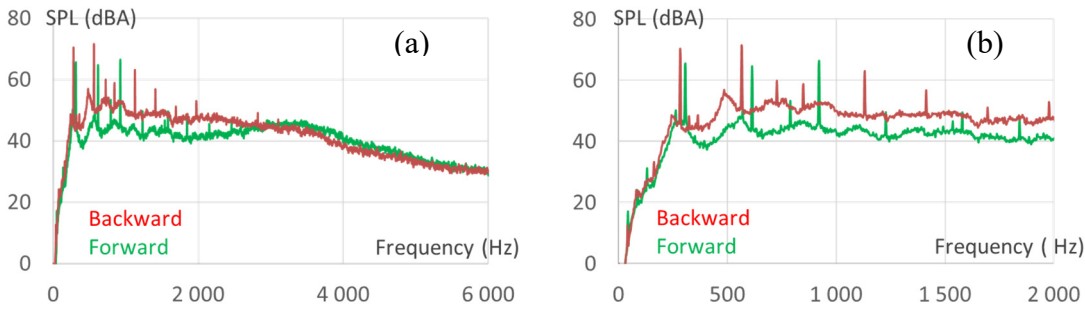

**Figure 9.** Compared acoustic spectra: (**a**) low to high frequencies, and (**b**) low to medium frequencies.

The frequency of blade passage and its harmonics are visible in the tonal peaks, with a higher level at BPF and H1 for the BW fan. The levels of the second and third harmonic are reversed between the front and rear blades, without, at this stage, a clear explanation.

In summary and as shown in the performance table, the backward curvature produces 3.6 dB more in the cooling module than the forward curvature (measured at the position of the microphone). 2.5 dB more was measured without exchangers: it indicates the same trend but it is however to be qualified because the acoustic test does not allow flow control (the fans are therefore operated at their zero pressure points).

## 5. Flow Analyses

The URANS numerical simulations were post-processed for the four configurations studied, and several effects are analyzed to explain the fan integration effects.

*5.1. Pressure Losses in the Heat Exchangers*

The very compact geometry of the cooling module does not allow for a homogeneous flow since the small space between the radiator and the nozzle creates areas of low speeds, especially in the corners and especially the two most distant corners. Velocity distribution on the inlet surface of the shroud for both cases with or without the exchanger is presented in Figure 10. The position of the fan is clearly visible through the circular footprint left on the speed distributions. Velocities are however smoother in the cooling module configuration, this being explained by the fact that the radiator resistance slows down the air in the central part and creates more suction in the corners.

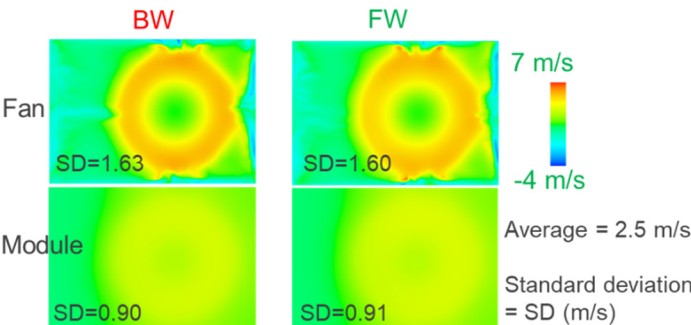

**Figure 10.** Axial velocity at the position of the heat exchanger. Comparison of the standard deviation with or without the heat exchanger.

The inhomogeneity of the flow is measured by comparing the deviation to the mean for each configuration. It is obtained by calculating the root mean square of the deviations from the average pondered by the surface. Standard deviations are in fact rather equivalent between the two fans: without the radiator, values are 1.63 and 1.60 m/s, respectively, for the backward and forward configurations, and 0.9 and 0.91 m/s, respectively, with the radiator. As these values are very close from one fan to the other, it is remarkable to note that both are showing a similar behavior in terms of flow distribution on the heat exchanger despite their strong geometrical differences. In fact, it can be assumed that the change of velocity given to the fluid by the fan is set by the same operating point, which explains the same flow rate distribution. The major effect is related to the presence or not of the exchanger, which promotes homogeneity by reducing the deviation from ~1.6 to ~0.9 m/s.

Numerical simulation allows us to evaluate the actual pressure drop in the heat exchanger using total pressure post-processing. The pressure drop through the radiator with an inhomogeneous flow distribution is numerically measured at 174 and 169 Pa (BW and FW, respectively), to be compared to the 140 theoretical Pa.

*5.2. Pressure Losses in the Shroud*

If all tests were conducted with the same shroud, it is conceivable that its effect would be different depending on whether or not it is equipped with radiators. Indeed, the presence of the exchanger forces the flow in the corners as shown in Figure 10, which increases friction in the small space between the radiator core and the shroud (separated by only ~36 mm). Post-processing of the total pressure losses between the surface of the core and the fan inlet gives values of −20 and 32 Pa (BW and FW, respectively), without giving any clear insight on the reasons for this small difference. In addition, it should be considered cautiously since the values cannot be considered as accurate since the post-processing involves surfaces close to the fan: the complexity of the flow with recirculation, large structure, etc. can alter the quality of the scalar averaging.

A summary of the pressure variations in the cooling module is presented in Table 2. It must be emphasized that finally the fans are producing a pressure rise which is slightly lower than the 206 and 231 Pa measured in the fan alone context.

**Table 2.** Post-processing of total pressure variations (Pa) in the cooling module.

|  | Radiator | Shroud | Fan |
|---|---|---|---|
| BW | −174 | −20 | 194 |
| FW | −169 | −32 | 201 |

## 5.3. Blade Loading Distribution

Velocity components are averaged in the azimutal direction and are shown in Figure 11. Axial velocities are relatively similar for the different cases, with, however, an acceleration of the flow velocity at 80% of the span for the backward fan in the module configuration. It should also be noted that the backward-curved fan produces an overall less tangential velocity and it is compensated by a largely positive radial velocity, whereas the forward curvature tends to create negative radial velocities. Some authors have already presented some effects of sweep or lean modifications [6–9,11,18], and it explains, for instance, that the blade curve creates a force acting on the flow that has a direction perpendicular to the surface of the pressure side. In the case of a forward fan, this force would be centripetal, and centrifugal in the case of a backward curvature.

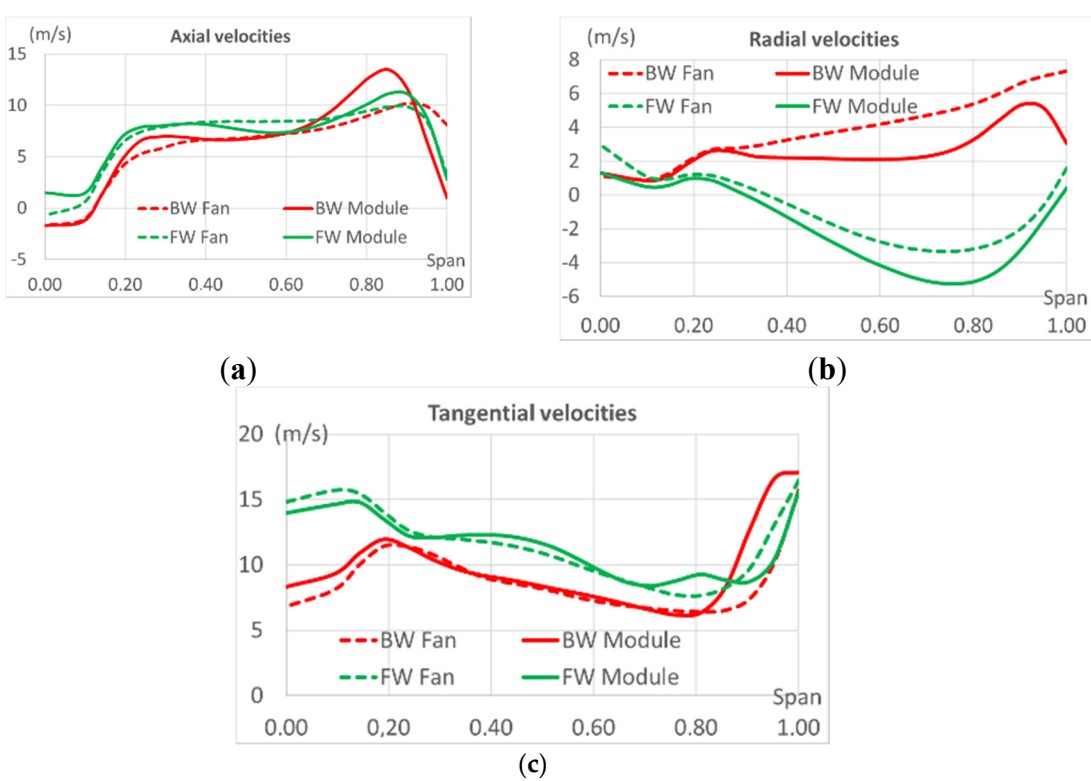

**Figure 11.** Velocity components for the different configurations: (**a**) axial velocity, (**b**) tangential velocity, and (**c**) radial velocity.

More radial flow as it exists for the backward fan tends to increase the load near the blade tip, roughly at 80% of the span (as described above). This is accompanied by a rather disturbed flow near the rotating ring, where more relative total pressure deficit is observed and presented with a circumferential averaging behind the trailing edge in Figure 12. Phenomena can be further analyzed in Figure 13, where it appears that a more important region of relative pressure loss is observed for the backward fan, nevertheless with a different pattern whether it is in the fan system (i.e., without the heat exchanger) or in the module configuration (i.e., with the heat exchanger). In the first case, the low pressure area extends upstream, while it goes on the internal surface of the rotating ring for

the second case. This is obviously the effect of the radiator resistance, which creates more suction effect on the side of the shroud and which pulls the area of losses towards the blade heads.

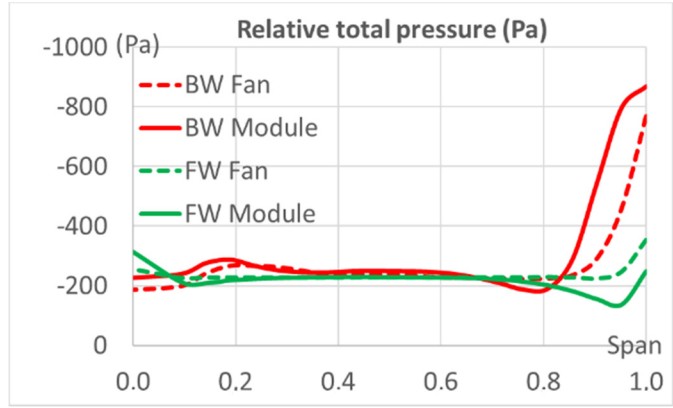

**Figure 12.** Deficit in relative pressure along the blade span (circumferential averaging 5 mm behind the trailing edge).

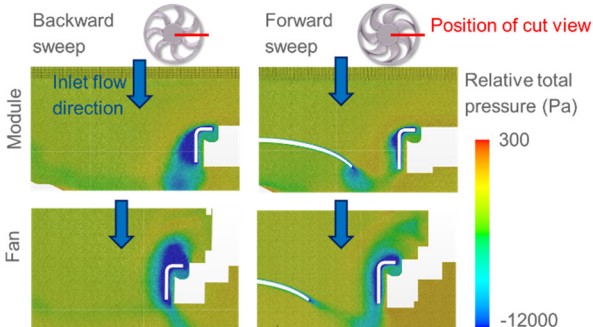

**Figure 13.** Cut view of the relative pressure distribution along the blade span.

If the losses at the tip of the blade are high for backward blades, a similarly disturbed area with a smaller radius appears for the forward curve. This can be seen in Figure 13 with the comparison of relative total pressure in cut planes selected between two consecutives blades, either in module configuration or with the fan alone. It has led to further analysis by examining the total relative pressure distribution around the blade profiles. The result presented in Figure 14 shows the extracted results at 80% and 95% of the span. It is clearly visible that the loss levels are very high at 95% for the backward fan (near the rotating ring) and a similar phenomenon is observed at 80% for the forward one. It can be hypothesized at this stage that the centripetal forces have brought the disturbance to a smaller radius than that of the backward curvature.

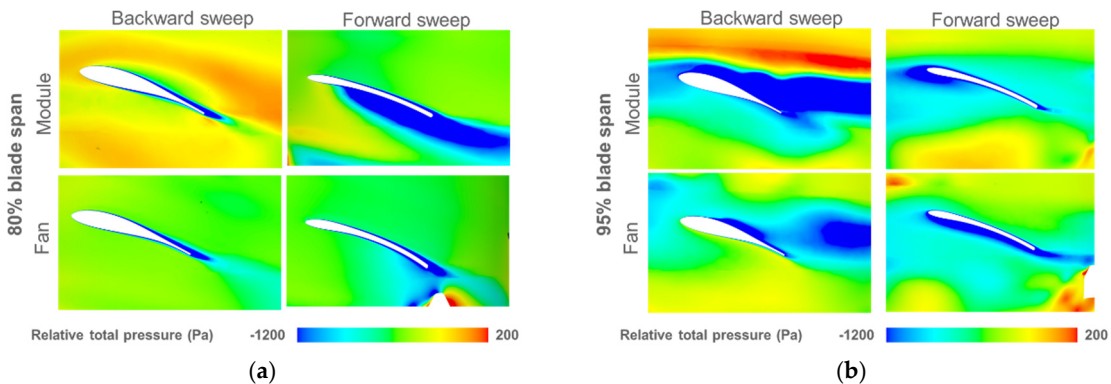

**Figure 14.** Relative total pressure distributions at constant span: (**a**) at 80% blade span, and (**b**) at 95% blade span.

These observations can be further visualized thanks to vorticity post-processing in module configuration for both fans. Figure 15 perfectly illustrates the creation of a flow structure winding around the periphery of the fan and more or less attached to the rotating ring. In the case of backward-curved blades, the blade heads see a highly turbulent flow on both the upper and lower surfaces, extending between 80% and 100% of the span. Blade heads are much less impacted by the presence of vorticity in the case of the forward curvature. However, there is confirmation of the existence of a structure passing over the pressure side and pushed towards the center of the fan, probably in connection with the centripetal flow. This turbulent winding interacts only with the inner face of the blade which is opposite to the position of the microphone used to acoustically characterize the module. It is at this stage rather difficult to determine if the blades can mask this acoustic source from the microphone: if the chord length can be considered acoustically compact (~5 cm), this is no longer the case if we consider the size of the fan (~400 mm). Further investigation to determine whether or not it can explain the acoustic difference between forward and backward curves is still needed.

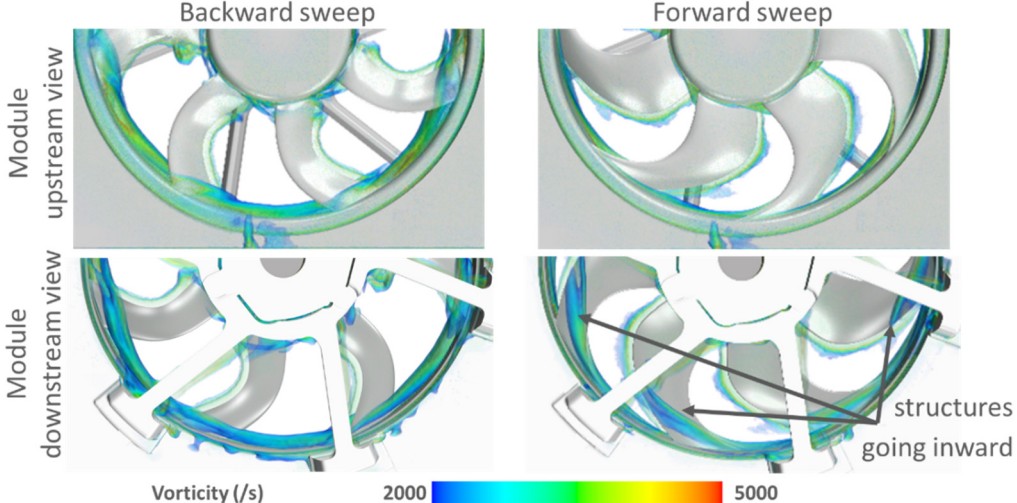

**Figure 15.** Vorticities in module configuration within the given range (upstream and downstream views): (**a**) backward curve, and (**b**) forward curve.

## 6. Acoustic Effects

### 6.1. Source Detection

As previously mentioned, the sound pressure level measured on the microphone is 3.6 dB smaller for the forward curve than for the backward curve. The previous vorticity post-treatments are the first elements making it possible to differentiate the noise generation mechanisms for the two fans, even if it is neither an acoustic prediction nor an identification of the sources.

However the effects of turbulence and the creation of structures in the flow contribute to generate pressure and density fluctuations on the rotating surfaces, and this is why it was considered useful to complete the study with DES-type numerical simulations. These calculations aim to record the pressure and density fluctuations on the walls in order to identify the dipoles, and possibly confirm the existing correlation between the source location and the presence of strongly disturbed flow. However, since the flow can create pressure fluctuations without effects on acoustics (a phenomenon called pseudo-sound), the post-processing was performed on density fluctuations to isolate sources more accurately.

A post-processing of the density on the walls at each time-step during the fan rotation highlights the spatial and temporal variations. It can be illustrated by a rate of change (here in kg/m$^3$/s) as in Figure 16, for a given time. It indicates qualitatively the main location of the sources at the top of the blade, and to a lesser extent at the blade bottom for the backward fan. The latter generally produces

more intense fluctuations, partially extending over the rotating ring. These observations are consistent with the hypothesis that flow disturbances and acoustic sources are related.

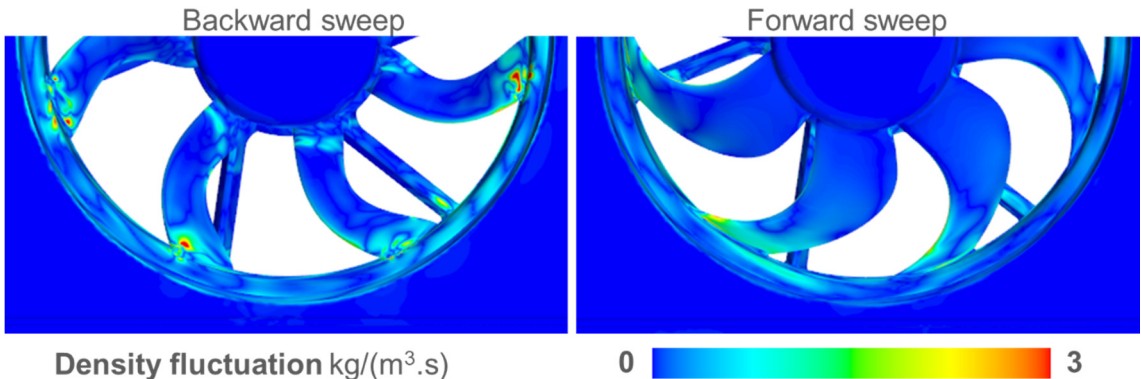

**Figure 16.** Comparison of density fluctuations on the fans.

*6.2. Propagation Effects*

After comparing the sources, attention can also be paid to the acoustic propagation. According to Amiet's theory [19] with dipolar sources, the main acoustic direction for a blade profile is perpendicular to the chord. In the case of a fan with complex blade shapes, this represents many directions depending on the geometry and the direction of propagation, which is more complex to define. It addition, the blade profile seen by the flow passing through a blade is no longer the geometric chord length and the various flow characteristics can lead to acoustic changes. It is therefore difficult to predict all the propagation phenomena, however the numerical simulation gives access to the FW&H analogy which allows measuring directivity effects. The comparison between the two fans in the module is presented both in Table 3 for the experimental-numerical comparison at the microphone position, and in Figure 17 for the directivity.

**Table 3.** Sound pressure level (dBA) at the position of the microphone.

|  | Experimental | Numerical |
| --- | --- | --- |
| BW | 80.3 | 83.1 |
| FW | 76.7 | 78.4 |
| Difference | 3.6 | 4.7 |

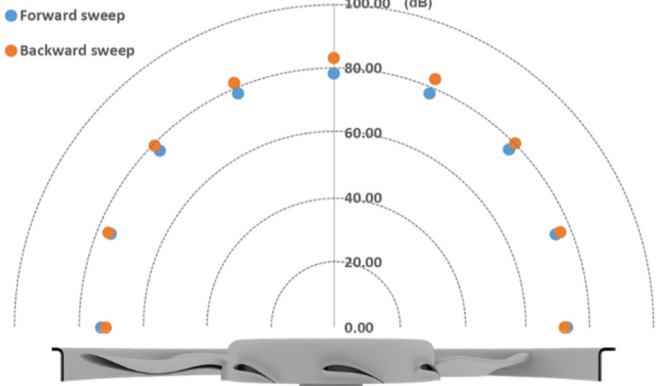

**Figure 17.** Directivity of the sound pressure level (dBA).

The simulations slightly over-predict acoustic levels. No explanation can be given to fully explain these differences, but it should be remembered that the numerical model does not take into account

the acoustic attenuation that may exist through the exchangers. However, it is observed that the trends are respected and that the calculation correctly predicts a lower acoustic level for the forward curvature.

The backward fan tends to emit more strongly on the axis, with 4.7 dB more than predicted numerically. The same calculations show that on the other hand, the propagation is weaker on the sides, since the difference is 1.4 dB in favor of the backward curved fan, which is the less noisy at this location.

## 7. Conclusions

A comparative study was undertaken between a forward and a backward curved fan. Tests and simulations confirm that the use of backward blades optimizes aerodynamic efficiency at the operating point at the expense of acoustics. It might be the effect of the blade loading which creates more radial velocity and then more change in momentum for the same torque.

From an acoustic point of view, the study confirms that the forward curvature has better acoustic properties, which can be explained by the smaller surface with intense dipolar sources at the blade tip and a lower propagation intensity in the direction of the axis of rotation (on which the measurement is made).

The difficulty for a correct design and dimensioning of an adapted fan lies in the fact that integration into the cooling module produces unpredictable effects. In particular, the adaptation of the operating point yields to flow rate and pressure clearly lower than those theoretically expected by considering the characteristic curves of exchangers and fans. Furthermore, unfortunately the variations are not necessarily of the same order of magnitude depending on whether one chooses a backward or a forward curved blade.

A fan optimization must therefore result from several considerations related to the physics of aerodynamics and aeroacoustics, but also related to the thermal and geometric constraints when integrated into a cooling module. The fan cannot therefore be designed alone, and a system approach is required.

**Author Contributions:** Prototypes and test organization: B.B.; Numerical simulation: B.D., M.L., M.A.; Numerical simulation with DES and acoustic post-processing: M.A.; Results Analysis: M.H., B.D., B.B.; Drafting the paper: M.H. Revising and approval of the paper: all authors. All authors have read and agreed to the published version of the manuscript.

**Funding:** This research received no external funding.

**Conflicts of Interest:** The authors declare no conflict of interest.

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
