# Peer review of "Effect of Blade Curvature on Fan Integration in Engine Cooling Module"

_acoustics, doi:10.3390/acoustics2040043_

Round 1

Reviewer 1 Report

Authors reported avery intresting study on fancurvature in engine cooling module. The approch herein described is very strong and scientific soundness. The paper mathc the requires for pubblication.

Nonetheless some minor issue are present.

Authors must add equation numbers in each equation reported in the text.

In section 5.1 authors reported the standard deviation. How was calulated? Which distribution was used? Is meaningfull? Discuss more in detail about the data distribution shown in this section.

Furthermore, authors reported figures alligned. In this case spilt them up or renumbered as figure xx a and figure xx b.

I recommand thi paper fro pubblication after minor revision.

Author Response

ANSWERS TO REVIEWER 1

Authors must add equation numbers in each equation reported in the text.

The equations have been numbered. It is true that given their small number we thought we could dispense ourselves with that.

In section 5.1 authors reported the standard deviation. How was calculated? Which distribution was used? It is meaningfull? Discuss more in detail about the data distribution shown in this section.

The calculation of the deviation uses the root mean square of the deviation from the mean over the measurement area. This has been added in the text, and the comments have been emphasized to show the little difference from one fan to another, and the notable difference with or without exchanger

The inhomogeneity of the flow is measured by comparing the deviation to the mean for each configuration. It is obtained by calculating the root mean square of the deviations from the average pondered by the surface. Standard deviations are in fact rather equivalent between the two fans: without radiator, values are 1.63m/s and 1.60 m/s respectively for backward and forward, and 0.9m/s and 0.91 m/s with radiator. As these values are very closed from one fan to the other, it is remarkable to note that both are showing a similar behavior in term of flow distribution on the heat exchanger despite their strong geometrical differences. In fact, it can be assumed that the change of velocity given to fluid by the fan is set by the same operating point, which explains the same flow rate distribution. The major effect is related to the presence or not of the exchanger, which promotes homogeneity by reducing the deviation from ~1.6 m/s to ~0.9 m/s.

Furthermore, authors reported figures alligned. In this case spilt them up or renumbered as figure xx a and figure xx b.

The way of presentation has been revised to be more formal and in line with the usage. We prefer to keep the figures with the same numbers, because each groups together several views to be compared between them. We adapt the format with the numbering "a", "b" and "c".

Reviewer 2 Report

A list of detailed comments is given below.

i)The paper suffers from insufficient description fan blades geometry, e.g. external fan diameter, blade chord etc.

Is a   pitch between neighbouring rotor blades on constant fan radius the same in peripheral direction?

ii) Measurement error of fan efficiency should be presented.

iii)Authors state in chapter no. 6, devoted to acoustic effects,  that the simulations slightly over-predict acoustic levels. Have authors compared this conclusion with published ones ?

After a careful revision of the paper by the authors, I recommend acceptance for journal  publication 

Author Response

ANSWERS TO REVIEWER 2

i)The paper suffers from insufficient description fan blades geometry, e.g. external fan diameter, blade chord etc.

Information giving the main dimensions of the fans has been provided in the newly proposed text.

Two fans are therefore considered in this study (Figure 2). They have been selected according to their curvature characteristics which are opposite to each other, i.e. a backward (BW) and a forward curvature (FW). Both have 7 blades equally distributed. The diameter at the blade tip is 423 mm, and 162 mm for the hub. The cords go from 42 mm to 59 mm on the backward fan, and from 67 mm to 86 mm on the forward fan. It is important to note that parameters such as chord, blade thickness and stagger angles have been intentionally modified to have fans with comparable performances. This could not have been the case otherwise since reversing the curvature leads to significant performance changes.

Is a   pitch between neighbouring rotor blades on constant fan radius the same in peripheral direction?

The blades are distributed equidistantly. This has also been clarified in the new text.

  1. ii) Measurement error of fan efficiency should be presented.

the curves come from numerical simulations. There is no random uncertainty associated with the calculation, and unfortunately the modeling error has not been the subject of a study to quantify the numerical uncertainty. This would be a considerable task, as we have seen in the simulation community, which has published extensively on the subject. This theme is unfortunately out of our reach if we refer to the activity it generates in congresses or publications.

However, the reader can appreciate the quality of the results by consulting reference 13 which reports on previous work on simulation-experiment comparison.

iii)Authors state in chapter no. 6, devoted to acoustic effects,  that the simulations slightly over-predict acoustic levels. Have authors compared this conclusion with published ones ?

There are very few works that report FW&H prediction from URANS or DES calculations for this type of fan, and to the knowledge of the authors none published with the consideration of the exchanger. All the authors acknowledge more or less significant discrepancies. Some results are presented in reference (6) for instance, these authors having also experimented the LBM methods. In any case, no precise explanation has been found in the literature to explain the numerical overprediction.

Given the state of the art in the field, and the lack of hindsight that the authors readily acknowledge, it is difficult to bring more elements into the context of this paper.